# Comparison of ICDAS, CAST, Nyvad’s Criteria, and WHO-DMFT for Caries Detection in a Sample of Italian Schoolchildren

**DOI:** 10.3390/ijerph16214120

**Published:** 2019-10-25

**Authors:** Guglielmo Campus, Fabio Cocco, Livia Ottolenghi, Maria Grazia Cagetti

**Affiliations:** 1Department of Restorative, Preventive and Pediatric Dentistry, University of Bern, Freiburgstrasse 7, 3010 Bern, Switzerland; 2Department of Surgery, Microsurgery and Medicine Sciences, School of Dentistry, University of Sassari, Viale San Pietro, 07100 Sassari, Italy; dr.fabiococco@gmail.com; 3Department of Oral and Maxillofacial Sciences, ‘Sapienza’ University of Rome, Via Caserta 6, 00161 Roma, Italy; livia.ottolenghi@uniroma1.it; 4Department of Biomedical, Surgical and Dental Science, University of Milan, Via Beldiletto 1, 20142 Milan, Italy; maria.cagetti@unimi.it

**Keywords:** dental caries, WHO-DMFT, ICDAS, CAST, Nyvad criteria

## Abstract

Caries measurement methods vary considerably in terms of the stages of lesion considered making the comparison problematic among different surveys. In this cross-sectional study, four caries measurement methods, the WHO-DMFT, the International Caries Detection and Assessment System (ICDAS), the Caries Assessment Spectrum and Treatment (CAST), and the Nyvad Criteria were tested in a sample of children. Five-hundred 12-year old children (236 males and 264 females) were examined four times by four calibrated examiners. The calibration process showed that Cohen’s Kappa exceeded the criterion of K = 0.75 and K = 0.80 for inter/intra-examiner agreement, respectively. In the survey, the total number of misclassification errors for the four methods amounted to 312 observations (67.94% regarding enamel lesions). The greatest difference among methods was shown by number of sound teeth (*p* < 0.01): WHO-DMFT *n* = 9505, 74.14%; ICDAS *n* = 2628, 20.49%; CAST *n* = 5053, 39.41%; and Nyvad Criteria *n* = 4117, 32.11%. At the level of dentinal Distinct/Active Cavity lesions, no statistically significant difference was observed (*p* = 0.40) between ICDAS (*n* = 1373, 10.71%), CAST (*n* = 1371, 0.69%), and Nyvad Criteria (*n* = 1720, 13.41%). In the severe caries levels, all methods were partially in agreement, while no accordance was found for the initial (enamel) lesions. A common language in caries detection is critical when different studies are compared.

## 1. Introduction

The fundamental purpose of any diagnostic procedure is to determine whether a subject has or does not have a particular condition. Although remarkable changes were observed about the prevalence and extent of oral health conditions, dental caries continues to be the most prevalent oral disease and one of the major public oral health issues [1,2,3]. These changes require modifications in preventive and therapeutic approaches, and the first step is to properly assess the presence and severity of the lesion. Recently [4], a consensus paper underlined the importance of caries severity assessment to plan preventive and/or therapeutic programs. A correct caries detection and classification contribute to identify caries risk patients, to detect enamel lesions, and to plan a non-operative treatment; moreover, in dentinal lesions, a correct classification provides indications about tissue preservation.

Methods to measure caries lesions are based on standardized diagnostic thresholds that allow a comparison of caries status and prevalence in different populations and countries worldwide.

In recent decades, a wide variety of new data collection methods have been developed to measure caries in individuals and groups; here, the most used and most recent ones will be briefly described (Table 1):

The Decayed–Missing–Filled (DMF) method proposed by WHO is the most common method in oral health epidemiology for assessing and measuring dental caries among populations. The method was developed more than 80 years ago [5,6]. When the method is operated in the permanent dentition, it is the sum of the number of teeth (interval 0–28) or surfaces (interval 0–128) that are decayed (D), missing (M), or filled (F) in an individual. The diagnostic threshold for the decayed tooth component (D) in the DMF is the cavitated dentine lesion [7]. This method, albeit having the advantage of being easy to apply, reaching high levels of reproducibility [7], excludes pre-cavitation stages from the measurement of the caries lesion.

The International Caries Detection and Assessment System (ICDAS) was developed in 2001 [8,9,10,11] with the aim to create a caries detection method that might be universally used and allowing clinicians, researchers, and epidemiologists to measure caries disease at different stages. The method was then updated as ICDAD II for coronal and root surface, and for caries assessment associated with restorations and sealants (CARS). The ICDAS is a two digit coding method; for caries, the method ranges from sound teeth (code 0), through enamel caries lesions (codes 1–3), to carious lesions in dentine (codes 4–6); for sealant and restoration, instead, the method ranges from 0 = Sound, 1 = Sealant, partial, 2 = Sealant, full, 3 = Tooth-colored restoration, 4 = Amalgam restoration, 5 = Stainless steel crown, 6 = Porcelain or gold or Porcelain-Fused-to-Metal (PFM) crown or veneer, 7 = Lost or broken restoration, 8 = Temporary restoration. Each surface is examined/coded, and when ICDAS is reported at tooth level, the worst condition is considered. More information about ICDAS is available from the website: http://www.icdas.org. Treatment needs are not considered in this method.

The Caries Assessment Spectrum and Treatment (CAST) instrument was developed with the goal to provide a valid reporting system combining both ICDAS II and WHO-DMFT methods [11,12,13]. The CAST method ranges from sound stage, through sealant, restoration, and to different stage of carious lesions (including lesion in enamel and dentine, progression in dental pulp and tooth-surrounding tissue, secondary lesions, as well as tooth lost due to caries); it can be used both at surface or tooth level. The code increases as the severity of the lesions due to the caries process increases.

The Nyvad Criteria were developed in 1999 [14,15]. They are based on a visual–tactile caries classification to enable the detection of the activity and severity of caries lesions, with special focus on low-caries populations, useful both in clinical practice and in a research setting. The caries process at surface or tooth level is classified into nine stages: Each severity stage from clinically sound surfaces/teeth through non-cavitated and micro-cavitated caries lesions in enamel, to frank cavitation into the dentine that can be classified in a double way, as active or inactive [16].

The DMFT, the ICDAS, and the CAST were recently compared in an adult population [16]. The DMFT, albeit being the fastest method to apply, had the disadvantage of underestimating the occurrence of lesions. The ICDAS, instead, recorded detailed information on caries severity through a high time-consuming measurement. Lastly, the CAST allowed to obtain information regarding disease distribution, lesion severity, and preventive/therapeutic needs at a time-rate similar to that of the DMFT. A limitation to these three methods is the absence of a validated definition of caries activity [16].

In clinical data assessment and analysis, it is crucial that the variables stating the disease (i.e., Caries) are registered with the least error possible. Often, the measurements obtained are error prone. When the variables under consideration are categorical, such error is termed misclassification error [17].

The visual–tactile examination is still essential in planning operative, non-operative, and epidemiological actions, and the number of caries lesions found depends on the diagnostic criteria and methods used; these criteria may vary considerably in terms of the stages of lesion considered. Even if all caries measurement methods have the ambition to be a universal model accepted for caries registration, their presence makes the comparison problematic among different surveys. Starting from these premises, a descriptive cross-sectional study was ideated, designed, and carried out, in which four caries measurement methods—namely the WHO-DMFT, the ICDAS, the CAST, and the Nyvad Criteria—were applied in a sample of schoolchildren.

## 2. Material and Methods

In 2016, an epidemiological survey called “National pathfinder on children’s oral health in Italy” was promoted by the Collaboration Centre for Epidemiology and Community Dentistry of Milan. This survey was the second National Survey conducted in Italy on children’s oral health. In 2016, the Italian population amounted to 60,589,445 people (29,445,741 males and 31,143,704 females) of whom 14.4% were younger than 15 years old. A multi-stage cluster sampling was performed, organizing Italy in sections according to the National Institute of Statistics: North-Western, North-Eastern, Central, Southern, and Insular Italy [18]. Secondary schools were chosen at cluster level with proportional random selection of participants for each of the counties identified in each section. A sample size for each stratum was calculated based on an assumed prevalence of dental caries (calculated using DMFT) of 43%, a standard error of 0.05, and a design effect of 2.5. A total of approximately 6000 Italian children attending the first year of secondary school was estimated for a final self-weighting sample. The size of a subsample was calculated using the same procedure, obtaining a minimum number of subjects (*n* = 427) with a power of 87%. The number of subjects enrolled was then increased up to the number of 500 children.

### 2.1. Training and Calibration of the Examiners

For the training, six examiners were provided with a guidance manual describing the four different caries detection methods and the respective examination criteria. Examiners had access to a collection of clinical photos illustrating caries criteria, as well as an explanation of the examination protocol. The examiner-trainees reviewed these materials independently and then, by successfully completing a minimum of 15 out of 20 questions (75%), passed a mandatory quiz.

As no true gold standard is generally available for caries diagnosis, a benchmark examiner may be used to assess sensitivity; in the present study, one of the authors (G.C.) acted as the benchmark validity reference for caries diagnosis. This author is a dentist who habitually uses all four caries detection methods and who had previously been trained and calibrated [19] in diagnosing caries lesions to ensure clinical registration’s comparability for epidemiological purposes [20].

The examiners attended a full-day course describing and discussing with the trainer the criteria of the different methods. Afterwards, all the examiners had to evaluate a second set of photo slides; four examiners that scored more than 90% were admitted to the clinical calibration exercise. The clinical calibration was performed in December 2016 in the Paediatric Dentistry Department of the Dental School, University of Sassari. Twenty-five children aged 12 years were examined and re-examined 10 times (twice for each examiner plus the gold standard) for each detection method during a two-week period. No discussion on the interpretation of the criteria was permitted between the examiners and the trainer.

### 2.2. Data Collection (Clinical Examinations)

The study duration was of five weeks from 16 January 2017 to 17 March 2017. The four examiners were calibrated (for detail see below). Each day of the survey, each examiner randomly selected a detection caries method and then the same bunch of children (*n* = 20) was examined. The procedure was repeated until all the children were examined according to all detection methods; in total, 2000 examinations were carried out and each child was examined four times.

For each evaluation, the four examiners, blinded to each other’s assessments, inspected every child. Before the first examination, children received a professional oral hygiene to remove calculus and plaque. The clinical examination was made under optimal lighting using a mirror and a World Health Organization probe to assess caries lesions. A compressed air syringe was used to dry the teeth during the application of the ICDAS method. The application time of each assessment method, from the first annotated code to the last recorded ones, was calculated. The mean time spent carrying out all examinations for each of the different methods was calculated and compared.

### 2.3. Caries Detection Methods

For the WHO-DMFT, the examiners recorded a tooth as decayed only if a cavity with detectably softened floor, undermined enamel, or a softened wall was detected; all caries stages that precede cavitation were considered sound [6,21].

For the ICDAS, the detection of caries was performed recording the two-digit codes for each tooth surface: The former, for tooth surface classification, choosing among sound, sealed, restored, crowned, or missing, and the latter, for the caries stage assessment, choosing among six scores, from sound to an extensive distinct cavity with visible dentine.

For the CAST, the examiners had to choose among ten codes: Sound, sealant, restoration presence, caries lesions in enamel or dentine, caries advanced stages involving pulp and tooth-surrounding tissue (abscess/fistula), and finally extracted teeth due to caries.

For the Nyvad Criteria, the examiners recorded the surface condition, choosing among nine scores: Sound, active/inactive caries with or without surface discontinuity or cavity, and filling with or without active/inactive caries lesion (Table 1).

### 2.4. Data Analysis

Examination outcomes were recorded in a spreadsheet (FileMaker Pro 9, FileMaker Inc., Santa Clara, CA, USA), an then imported to a statistical software program (STATA 13 for Mac, STATACorp., College Station, TX, USA). Descriptive statistics (absolute counts, percentage, mean, and standard deviation) were calculated for each method. The tooth was considered the unit for all analyses. When the method supports the registration at the surface level, the maximum value recorded per tooth was considered.

Inter-examiner reliability was evaluated through the analysis of variance for fixed effect [19]. The strength of agreement associated with Kappa statistics was labeled as <0.51 slight, 0.51–0.60 fair, 0.61–0.70 acceptable, 0.71–0.80 moderate, 0.81–0.90 substantial, >0.90 almost perfect [22,23]. Kappa statistics were tested through z test at a significance level of 0.01. Misclassification errors were recorded and analyzed, and the percentage distribution of misclassified observations was calculated.

Moreover, the agreement was also calculated at grey shadow/dentine level among the ICDAS and the CAST.

## 3. Results

The sample consisted of 500 children (52.80% females and 47.20% males mean age in year 11.62 ± 0.65, age range 10.9–13.02 yy). Vital statistics (i.e., gender, educational level of the family, working status of the parents) were quite homogeneous; the majority of the children had a mother born in the European Union with a compulsory educational level and working as clerks or self-employed (Table 2).

### 3.1. Calibration

Table 3 shows the inter-examiner agreement between the benchmark and the four examiners and the intra-examiners calibration in the two sessions, both for sound teeth and distinct caries lesion stage. For each detection method, a good agreement between the examiners and the benchmark was recorded, even if the WHO-DMFT showed the highest k value, both for sound (K-Cohen between 0.83 and 0.92) and distinct caries (K-Cohen between 0.77 and 0.89), and the ICDAS the lowest (K-Cohen for sound between 0.72 and 0.85; for distinct caries between 0.75 and 0.85). Similarly, the intra-examiner agreement, reaching the highest k values for the Nyvad Criteria (between 0.91 and 0.96 for sound teeth and 0.84 and 0.90 for distinct caries), was also good.

In regard to grey shadow/dentine lesion level, a good inter-examiner reproducibility was observed both for the ICDAS and the CAST (K-Cohen 0.82 and 0.83, respectively), data not in table.

Inter-examiner reliability via analysis of variance is reported in Table 4. In Session 1 of the calibration, a good agreement between examiners for all methods, without significant differences among them and with a *p*-value ranging from 0.10 for the ICDAS to 0.17 for the WHO-DMFT at the sound level and from 0.12 for the ICDAS to 0.14 for the CAST at the distinct caries level, was found. In Session 2, the *p*-value increased drastically for all methods.

### 3.2. Survey

Table 5 and Table 6 report the classification of the tooth conditions according to the four different methods and the accordance among methods.

The total number of misclassification errors (Table 5) for the four methods amounted to 312; 13 (1.86%) for WHO-DMFT, 125 (17.86%) for ICDAS, 83 (11.86%) for CAST, and 91 (13.00%) for Nyvad Criteria. The majority of misclassifications involved enamel lesions (i.e., enamel opacity and enamel opacity wet for the ICDAS, enamel discontinuity for the CAST, and activity no cavity lesions for the Nyvad Criteria). Using the WHO-DMFT, the errors related to sound and distinct caries amounted to 7 teeth. Using the ICDAS, the disallocation errors related to sound and enamel caries (enamel opacity/enamel opacity wet/enamel discontinuity) amounted to 23 teeth, while related to enamel caries and dentine caries (grey shadow/distinct caries/pulp involvement) amounted to 22 teeth, and lastly, ascribed to sound and dentine caries amounted to 1 tooth. For the CAST, the misclassification errors related to sound and enamel caries (enamel discontinuity) amounted to 20 teeth, related to enamel caries and dentine caries (grey shadow/distinct cavity/pulp involvement) amounted to 8 teeth, and finally, no differences were found assigned to sound and dentine caries. Considering the Nyvad Criteria, the disallocation errors related to sound and no cavity lesions (activity no cavity/inactive no cavity/active discontinuity/inactive discontinuity) amounted to 12 teeth and ascribed to cavity and no cavity lesions amounted to 3 teeth; no differences were found related to sound and cavity lesions, and lastly, errors assigned to active and inactive lesions amounted to 27 teeth. Regarding the application time, the fastest method was the WHO-DMFT, with a mean application time of 3.7 ±1.2 min, while for the ICDAS it was 6.3 ± 3.6 min, for the CAST 5.2 ± 4.2 min, and for the Nyvad Criteria 5.1 ± 3.5 min (data not in tables). 

The sample distribution regarding tooth conditions according to the WHO-DMFT, ICDAS, CAST, and Nyvad Criteria methods is shown in Table 6. The percentage of sound teeth recorded using the four methods was statistically significant different (^2^_(3)_ = 83.0, *p* < 0.01), with the WHO-DMFT showing the highest value (74.14%) and the ICDAS the lowest (20.49%). Data regarding distinct/active lesions were also compared among the WHO-DMFT, the ICDAS, the CAST, and the Nyvad Criteria; the percentages measured using the four methods were statistically significantly different (^2^_(3)_ = 224.05, *p* < 0.01), with the ICDAS and the CAST showing the lowest (10.71% and 10.69%, respectively) and the WHO-DMFT the highest (22.45%). No statistically significant differences were observed for the other severity stages considered. Data about fillings and missing teeth for caries did not show any significant difference; only the sealant’s presence differed among the three methods reporting sealant data, namely WHO-DMFT, ICDAS, CAST (^2^_(2)_ = 48.08, *p* < 0.01).

## 4. Discussion

The process of choosing the best model to evaluate a disease is a trade-off between simplicity and accuracy. This is particularly true for caries disease in childhood, since a large proportion of subjects have no caries in dentine. To try to solve these problems, several measurement methods other than DMFT were created [24]. Caries assessment methods have the goal of evaluating and recording consistent and standardized data of tooth condition, providing information that might be used for clinical, research, and epidemiological purposes.

An important aspect of any study is the use of appropriate methodologies either to control or to reduce the effects of potential confounding factors (i.e., the comparison of data from different surveys). An element able to severely influence the outcomes of dental caries in scientific studies is the variation in disease diagnosis among different methods, making the comparison between surveys almost impossible. Carious lesion management is nowadays based on a non-operative manner [4,25]; the choice of a criterion of caries detection that includes non-cavitated lesions and early injuries could increase sensitivity, mainly in populations with low prevalence of the disease.

The present study aimed to compare the caries data recorded in a sample of schoolchildren aged 12 years using four methods: The WHO-DMFT, the ICDAS, the CAST, and the Nyvad criteria. Although these methods were developed for different goals in distinct historical periods and they present different strengths and weaknesses, they are frequently used in a similar context.

Its simplicity and limited time-consuming application make the WHO-DMFT the most used and preferable method in large epidemiological surveys or in operative treatment needs evaluations [26]. Moreover, whilst this method is suitable for estimating disease prevalence and incidence in an adult population, in children and the elderly, a relevant amount of information is likely to be lost, possibly leading to an underestimation of caries in these populations [27]. This was confirmed in the present survey, where about two thirds of the teeth using DMFT had been recorded as sound.

The ICDAS method has several benefits, including a high accuracy, since coding the lesion’s diverse stages helps clinicians and researchers to differentiate the different stages of the disease. As reported above, however, the ICDAS has a lower reproducibility compared to the WHO-DMFT, the CAST, and the Nyvad Criteria. This limitation was also recorded in this survey, since when using the ICDAS, a higher number of misclassification errors was found compared to the other methods.

The CAST can be used to assess caries whilst the evaluation of the need of surgical treatment is conducted. This method classifies caries lesions in hierarchical order according to their severity, including missed teeth due to caries. A limitation of the CAST observed in this survey was the total absence regarding the activity of the lesions.

In high risk subjects, the Nyvad Criteria method might be useful in order to control the activity of the disease, helping clinicians to choose the most appropriate treatment plan; moreover, the method might be helpful for planning and evaluating population-based preventive programs. The Nyvad Criteria in the present survey proved its time-consuming nature and lack of missing teeth due to caries.

Good Kappa values regarding inter- and intra-reliability scores among the four examiners were recorded. These results underline that even more exhaustive assessment methods, i.e., grading from enamel to dentine lesions and various codes for lesion activity and fillings, do not diminish reliability when a good calibration of the examiners is performed. Otherwise, the Kappa value is dependent on the diagnosis of the unknown condition prevalence [12,27]. In this paper, albeit all assessment methods recorded different caries registrations, Kappa values had a low variation among the methods, as reported in the results. In the calibration, the four examiners were trained and tested to score the WHO-DMFT, the ICDAS, the CAST, and the Nyvad Criteria. A proof of the simplicity of the WHO-DMFT is the higher agreement among the four examiners in detecting both the sound tooth and the distinct caries already in the first calibration session, compared to the other methods. The intra-examiner agreement evaluation shows that the CAST and the Nyvad Criteria are the methods that reach the highest agreement, allowing each examiner to be more consistent with his own judgment.

Comparing the four methods is not easy, since they include different scores regarding both the lesion stage and the restoration adopted; even the scores included in all the methods are not fully comparable. For instance, regarding the absence of the lesion (sound tooth), a score included in all methods, data show a huge variability: The WHO-DMFT records about two-thirds of the sample as caries-free, whilst using the ICDAS, only a fifth of the teeth was recorded as sound; the CAST and the Nyvad Criteria showed intermediate values, more similar to each other. This result was foreseeable, as the WHO-DMFT considers early stages of the lesions, i.e., enamel and early dentine lesions, as sound, while they are considered as affected in the other methods [13,28].

In the survey, the misclassification error percentages are almost insignificant for the WHO-DMFT method, although the misclassification leads to judging a sound tooth as carious or the opposite. For the other three methods, the misclassification error percentages were found mainly for enamel and no cavity caries lesions, showing how difficult the clinical detection of the early stages of the caries process is. One of the possible reasons for the misclassification errors might be a certain conceptual deadlock of the methods (i.e., to differentiate between grey shadow, enamel opacity, enamel discontinuity, and so on). Accordance was observed only for the most severe caries levels among the different detection methods, while initial caries levels, namely enamel lesions, showed almost no accordance at all.

The planning of a non-operative caries treatment at the early stages of the lesion or of an operative caries treatment at more severe stages (i.e., cavity in dentine) are nowadays considered the best clinical practices in dentistry. Therefore, assessing the lesion only at a dentinal cavitated stage (the WHO-DMFT method) precludes the possibility of non-operative care of the disease.

No method allows to estimate the caries progression rate yet; nevertheless, the activity of the lesion, included in Nyvad Criteria [29], might be considered as a proxy of the future progression of the lesion, since it reflects the demineralization activity in the dental biofilm.

In Italy, national dental health care is almost completely private-based, which might explain the low percentages of teeth with fillings. In the last National Survey of oral health in children, the restored component represented only a fraction of the examined teeth [30].

The main limitation of this survey is related to the population enrolled, schoolchildren with a rather low caries prevalence, and one may, therefore, expect the results to apply only in low-caries populations. Caries detection methods among adults, the middle-aged, and the elderly may be compounded by their often much greater restorative experience, and the results of the present study may not necessarily be broadened to those populations.

Another hypothetical limitation of this study lays in the calibration process itself. In theory, calibration is based on the assumption that only true scores are recorded by a gold standard (instrument or examiner) that is theoretically 100% error-free. In clinical oral settings, the scores are generated by a benchmark scorer, usually an experienced examiner who is assumed to be error-free or nearly so, but, of course, some misclassification errors are expected.

On the other hand, one of the strengths of the present survey is the wide sample included; to the authors’ knowledge, no other study has included a so complete calibration and reliability analysis providing credit to the external validity of the findings.

This paper might be of primary interest of clinicians, epidemiologists, and dental researchers.

Clinicians have to select the caries detection method that best fits with their daily outcome routine (especially for non-operative treatments). Furthermore, they have to know the different methods when reporting their outcomes or reading literature data.

Epidemiologists might find this paper of interest as, for the first time, a complete palette of the performance of the most recent caries detection methods is presented. They have to be able to decide which method they have to select, taking into account the goals for their surveys.

Dental researchers would also find this paper relevant; like the epidemiologists, they have to select the best detection methods in relation to the aims and outcomes to correct plan trials.

## 5. Conclusions

The outcomes of the present survey allowed to draw these conclusions:A certain grade of accordance among all the methods was found for severe caries levels, while no accordance for the initial (enamel) lesions.From a clinical, epidemiological, and research prospective, both the severity and the activity of a caries lesion are important factors to consider.A common language in caries detection is crucial when different studies are compared.

## Figures and Tables

**Table 1 ijerph-16-04120-t001:** Hierarchy Clinical Code scores methods.

Lesion Stage	WHO-DMFT	ICDAS	CAST	Nyvad Criteria
No lesion (sound)				
First visual change in enamel (dry)				
Distinct visual change in enamel				
Active intact surface				
Inactive intact surface				
Enamel discontinuity				
Integrity loss/Active discontinuity				
Grey shadow/Dentine	+			
Distinct/Active cavity	+			
Extensive distinct cavity				
Inactive caries discontinuity				
Inactive cavity				
Pulp involvement				
Abscess/Fistula				
Missing tooth for caries				
Missing tooth for other reason				
Unerupted				
Sealant		^		
Filling (sound surface)		*		
Filling + active caries				
Filling + inactive caries				
Temporary filling				

+ for DMFT, different cut-off points for caries diagnosis were reported in different surveys, using the grey shadow in dentin or the distinct cavity. * for ICDAS, including filling tooth-colored, amalgam, stainless still crown, Porcelain or gold, or PFM crown or veneer. ^ for ICDAS, including sealant, partial, and full. WHO-DMFT, WHO Decayed–Missing–Filled Teeth; ICDAS, International Caries Detection and Assessment System; CAST, Caries Assessment Spectrum and Treatment.

**Table 2 ijerph-16-04120-t002:** Childrens’ socio-behavioral demographic characteristics of the sample by gender.

Measure	Samples
	***n* (%)**
**Gender**	
Males	236 (47.20)
Females	264 (52.80)
**Maternal Nationality**	
European Union	455 (91.00)
Not European Union	45 (9.00)
**Educational level of the mother**	
Compulsory education	224 (44.80)
Secondary school	121 (24.20)
University	155 (31.00)
**Educational level of the father**	
Compulsory education	242 (48.40)
Secondary school	141 (28.20)
University	117 (23.40)
**Occupational status of the mother**	
Housewife	126 (25.20)
Unemployed	102 (20.40)
Clerks	131 (26.20)
Self-employed	141 (28.20)
**Frequency of toothbrushing**	
1/day	28 (5.60)
2/day	73 (14.60)
>2 day	399 (79.80)

**Table 3 ijerph-16-04120-t003:** Calibration agreement (criterion K = 0.75) of the four examiners (A–D) vs. the benchmark for the different caries detection methods and intra-examiner agreement (criterion K = 0.80). The K-Cohen value was calculated at the sound level and at the distinct caries level; moreover, a total value was calculated.

		Inter-Examiner Agreement vs. Benchmark						Intra-Examiner Calibration		
		Session 1			Session 2			Session1/Session 2		
Methods	Examiner	K-Cohen	K-Cohen	K-Cohen	K-Cohen	K-Cohen	K-Cohen	K-Cohen	K-Cohen	K-Cohen
		(Sound)	(Distinct Caries)	(Total)	(Sound)	(Distinct Caries)	(Total)	(Sound)	(Distinct Caries)	(Total)
WHO-DMFT	A	0.83	0.81	0.87	0.91	0.87	0.88	0.94	0.83	0.84
	B	0.84	0.77	0.81	0.86	0.84	0.85	0.89	0.80	0.83
	C	0.89	0.81	0.86	0.92	0.86	0.88	0.91	0.84	0.85
	D	0.84	0.79	0.82	0.83	0.89	0.87	0.90	0.88	0.89

ICDAS	A	0.74	0.75	0.76	0.77	0.80	0.79	0.84	0.83	0.83
	B	0.79	0.76	0.77	0.81	0.80	0.80	0.87	0.89	0.88
	C	0.76	0.84	0.80	0.85	0.85	0.85	0.91	0.88	0.89
	D	0.72	0.77	0.74	0.74	0.82	0.78	0.87	0.86	0.86

CAST	A	0.73	0.72	0.72	0.81	0.83	0.82	0.87	0.84	0.86
	B	0.74	0.76	0.75	0.82	0.85	0.84	0.85	0.82	0.83
	C	0.82	0.78	0.80	0.83	0.90	0.86	0.92	0.89	0.90
	D	0.74	0.75	0.74	0.81	0.83	0.82	0.94	0.88	0.91

Nyvad Criteria	A	0.71	0.73	0.72	0.76	0.81	0.79	0.91	0.84	0.87
	B	0.77	0.74	0.75	0.80	0.82	0.81	0.92	0.88	0.90
	C	0.81	0.78	0.80	0.83	0.87	0.85	0.92	0.87	0.90
	D	0.79	0.76	0.77	0.85	0.83	0.84	0.96	0.90	0.94

**Table 4 ijerph-16-04120-t004:** Inter-examiner reliability: Analysis of variance on the four examiners using the different methods.

			WHO-DMFT	ICDAS	CAST	Nyvad Criteria
Sound	Session 1	Sum of squares	3971.44	3428.16	3562.56	3629.17
		F ratio	104.27	93.63	98.18	99.25
		*p*-value	0.17	0.10	0.13	0.16
	Session 2	Sum of squares	4217.84	3823.54	3924.34	4202.71
		F ratio	448.34	203.62	336.19	442.74
		*p*-value	0.21	0.16	0.16	0.17
Distinct caries	Session 1	Sum of squares	376.220	3644.71	3802.57	3784.71
		F ratio	100.41	97.85	102.72	101.57
		*p*-value	0.13	0.12	0.14	0.13
	Session 2	Sum of squares	428.331	4072.56	4188.67	4266.51
		F ratio	477.53	284.52	432.32	461.74
		*p*-value	0.17	0.16	0.17	0.17

**Table ijerph-16-04120-t005a:** (**A**)

WHO-DMFT	Sound	Distinct Cavity	Filling	Missing Tooth	Sealant	Total	Misclassification
							*N* %
**Sound**	**563**	2	0	0	0	*565*	*5 (0.89)*
**Distinct cavity**	5	**90**	0	0	0	*95*	*5 (5.55)*
**Filling**	0	0	**19**	0	1	*20*	*1 (5.26)*
**Missing tooth**	0	0	0	**1**	0	*1*	*-*
**Sealant**	0	1	1	0	**17**	*19*	*2 (7.14)*
**Total**	*568*	*93*	*20*	*1*	*18*	*700*	*13 (1.86)*

**Table ijerph-16-04120-t005b:** (**B**)

ICDAS	Sound	Enamel Opacity	Enamel Opacity Wet	Integrity Loss	Grey Shadow	Distinct Cavity	Pulp Involv.	Filling	Missing Tooth	Sealant	Total	Misclassification N (%)
**Sound**	**145**	12	2	1	0	0	0	0	0	3	*163*	*18 (12.41)*
**Enamel opacity**	11	**93**	13	0	4	0	0	0	0	0	*121*	*28 (30.11)*
**Enamel opacity wet**	2	9	**111**	1	9	0	0	0	0	0	*132*	*23 (20.72)*
**Integrity loss**	0	0	2	**65**	3	0	0	0	0	0	*70*	*12 (18.46)*
**Grey shadow**	1	2	9	0	**47**	6	0	0	0	0	*65*	*22(46.81)*
**Distinct cavity**	0	0	0	10	6	**83**	2	0	0	0	*101*	*18 (21.69)*
**Pulp involvement**	0	0	0	0	0	0	**9**	0	0	0	*9*	-
**Filling**	0	0	0	0	0	0	0	**19**	0	1	*20*	*1 (5.26)*
**Missing tooth**	0	0	0	0	0	0	0	0	**1**	0	*19*	-
**Sealant**	0	0	0	0	0	0	0	0	0	**18**	*19*	*3 (16.67)*
**Total**	*159*	*116*	*137*	*77*	*70*	*89*	*11*	*19*	*1*	*22*	*700*	*125 *(17.86)

**Table ijerph-16-04120-t005c:** (**C**)

CAST	Sound	Enamel Discontinuity	Grey Shadow	Distinct Cavity	Pulp Involv.	Filling	Missing Tooth	Sealant	Total	Misclassification N (%)
**Sound**	**245**	12	0	0	0	0	0	4	*261*	*20 (8.16)*
**Enamel discontinuity**	20	**178**	5	3	0	0	0	0	*206*	*24 (13.48)*
**Grey shadow**	0	10	**67**	0	0	0	0	0	*77*	*20 (20.91)*
**Distinct cavity**	0	2	9	**95**	1	0	0	0	*107*	*12 (12.63)*
**Pulp involvement**	0	0	0	1	**9**	0	0	0	*10*	*1 (11.11)*
**Filling**	0	0	0	0	0	**19**	0	1	*20*	*1 (5.26)*
**Missing tooth**	0	0	0	0	0	0	**1**	0	*1*	-
**Sealant**	0	0	0	0	0	0	0	**18**	*18*	*5 (21.73)*
**Total**	*265*	*202*	*81*	*99*	*10*	*19*	*1*	*23*	*700*	*83 (11.86)*

**Table ijerph-16-04120-t005d:** (**D**)

NYVAD	Sound	Activity No Cavity	Inactive No Cavity	Active Discontinuity	Active Cavity	Inactive Discontinuity	Inactive Cavity	Total	Misclassification N (%)
**Sound**	**169**	0	12	0	0	0	0	*181*	*12 (7.10)*
**Activity no cavity**	0	**91**	8	3	0	0	0	*102*	*19 (20.90)*
**Inactive no cavity**	9	15	**82**	1	0	0	0	*107*	*25 (30:49)*
**Active discontinuity**	0	4	3	**75**	1	0	0	*83*	*8 (10.67)*
**Active cavity**	0	0	0	3	**113**	0	10	*126*	*13 (11.50)*
**Inactive discontinuity**	2	0	0	0	0	**16**	0	*18*	*2 (12.50)*
**Inactive cavity**	0	0	0	0	12	0	**33**	*45*	*12 (36.36)*
**Total**	*180*	*110*	*105*	*82*	*126*	*16*	*43*	*662*	*91 (13.00)*

**Table 6 ijerph-16-04120-t006:** Accordance among the WHO-DMFT, ICDAS, CAST, and Nyvad Criteria methods related to tooth condition. Percentage of Accordance error refers to the columns.

Tooth Conditions	WHO-DMFT *(n = 12,820)* *n* (%)	ICDAS *(n = 12,824)* *n* (%)	CAST *(n = 12,822)* *n* (%)	Nyvad Criteria *(n = 12,824)* *n* (%)	Statistical Analysis
No lesion (Sound)	9505 (74.14)	2628 (20.49)	5053 (39.41)	4117 (32.11)	χ^2^_(3)_ = 83.0 *p* < 0.01
Enamel opacity		1782 (13.90)			
Activity no cavity				2896 (22.59)	
Inactive no cavity				1398 (10.90)	
Enamel opacity wet		2340 (18.25)			
Enamel			4011 (31.28)		
Integrity loss/Active discontinuity		2241 (17.48)		1430 (11.15)	χ^2^_(1)_ = 1.72 *p* = 0.19
Grey shadow/Dentine		1229 (9.58)	1156 (9.02)		χ^2^_(1)_ = 0.07 *p* = 0.84
Distinct/Active cavity	2878 (22.45)	1373 (10.71)	1371 (10.69)	1720 (13.41)	χ^2^_(2)_ = 224.05 *p* < 0.01
Inactive discontinuity				832 (6.49)	
Inactive cavity				431 (3.36)	
Pulp involvement		797 (6.22)	796 (6.21)		------------
Filling	425 (3.32)	420 (3.28)	423 (3.30)		χ^2^_(2)_ = 0.04 *p* = 0.99
Missing tooth for caries	12 (0.09)	12 (0.09)	12 (0.09)		----------
Sealants	224 (1.75)	365 (2.85)	223 (1.74)		χ^2^_(2)_ = 48.08 *p* < 0.01

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
