# Peer review of "Comparison of ICDAS, CAST, Nyvad’s Criteria, and WHO-DMFT for Caries Detection in a Sample of Italian Schoolchildren"

_ijerph, 2019, doi:10.3390/ijerph16214120_

Round 1
Reviewer 1 Report
I believe you need a better description of why this study was done and why it is important. These classification systems were designed for different purposes, so therefore wouldn't be expected to be comparable. Are you not "comparing apples to oranges?"
Throughout you have lumped clinical, epidemiological, and research uses together. This isn't logical and makes your introduction, discussion, and conclusion confusing. Line 39 of the introduction states "these changes require modifications in preventive and therapeutic approaches." But the caries classification methods are developed for screening (epidemiology) and research. This confusion continues throughout.
In the introduction, it would be helpful to have a paragraph for each method to describe all the details about it, including the purpose for which it was created. As written, the information in jumbled together in one paragraph.
What is the actual age range of your "12 year olds." Are they all actually 12?
Two methods collecting caries information by surface and two by tooth. How are these compared? Did you reduce the ICDAS and THe Nyvad to tooth level? Select the worst surface to compare to the tooth in the others?
How data collection occurred is not clear. Did each of the 500 children receive four exams by each of the four examiners, for a total of 16 examinations each? Or did each examiner see only one quarter of all the children?
Why are children compared by gender? It doesn't seem relevant to this research unless you also report on caries by gender.
Should intra- and inter examiner reliability be published separately. This seems to be the most important part of the current paper.
The paragraph in the results that begins on line 172 is very confusing. Could this be presented in a table format more clearly? A little more information about what these results mean would also be helpful.
Could you a priori group the classifications in IDCAS the would represent sound and decayed so that it could be more closely compared to DMFT? As written, the discussion and conclusion do not convince me that the comparison is important. Could you add more about what is learned about each classification system from your research and that means for readers?
Author Response
First of all, I in behalf of all the authors want to thank to the reviewer. The comments were really helpful to modify and we hope to improve the paper.
In details:
Reviewer comments:
I believe you need a better description of why this study was done and why it is important. These classification systems were designed for different purposes, so therefore wouldn't be expected to be comparable. Are you not "comparing apples to oranges?"
Our Reply:
We completely re-written the introduction part following the reviewer suggestion. We stressed how the study is important. “Since the visual–tactile examination is still essential in planning operative, non-operative and epidemiological actions, the number of caries lesions found depends on the diagnostic criteria and methods used; these criteria may vary considerably in terms of the stages of lesion considered. Even if, all caries measurement methods have the ambition to be a universal model accepted for caries registration, their presence makes the comparison problematic among different surveys. Starting from these premises, it was ideated, designed and carried out a descriptive cross-sectional study, in which four caries measurement methods, namely the WHO-DMFT, the ICDAS, the CAST and the Nyvad Criteria were applied in a sample of schoolchildren.”
Reviewer comments:
Throughout you have lumped clinical, epidemiological, and research uses together. This isn't logical and makes your introduction, discussion, and conclusion confusing. Line 39 of the introduction states "these changes require modifications in preventive and therapeutic approaches." But the caries classification methods are developed for screening (epidemiology) and research. This confusion continues throughout.
Our Reply:
We thank the reviewer for the comment, but we partially disagree to the comment. Caries classification is of fundamental importance to plan preventive or therapeutic approaches. Just this week a new consensus was published and this is the definition on Caries classification and diagnosis “Caries diagnosis is the clinical judgement integrating available information, including the detection and assessment of caries signs (lesions), to determine presence of the disease. The main purpose of clinical caries diagnosis is to achieve the best health outcome for the patient by selecting the best management option for each lesion type, to inform the patient, and to monitor the clinical course of the disease.”
Reviewer comments:
In the introduction, it would be helpful to have a paragraph for each method to describe all the details about it, including the purpose for which it was created. As written, the information in jumbled together in one paragraph.
Our Reply:
We completely revised the introduction section following reviewer’s suggestions. Now each method is described in a separate paragraph.
Reviewer comments:
What is the actual age range of your "12 year olds." Are they all actually 12?
Our Reply:
The sample consisted of 500 children (52.80% females and 47.20% males mean age in year 11.62±0.65, age range 10.9 – 13.02 yy).
Reviewer comments:
Two methods collecting caries information by surface and two by tooth. How are these compared? Did you reduce the ICDAS and The Nyvad to tooth level? Select the worst surface to compare to the tooth in the others?
Our Reply:
As is now written in data analysis “When, the method supports the registration at the surface level, the maximum value recorded per tooth was considered”.
Reviewer comments:
How data collection occurred is not clear. Did each of the 500 children receive four exams by each of the four examiners, for a total of 16 examinations each? Or did each examiner see only one quarter of all the children?
Our Reply:
We clarified that each child was examined four times by four examiners; so, in total 2000 examinations were carried out.
Reviewer comments:
Why are children compared by gender? It doesn't seem relevant to this research unless you also report on caries by gender.
Our Reply:
No data related to the survey is provided by gender; we used gender as strata only to describe the sample. Anyway, we followed the reviewer’s suggestions; so, we modified table 2.
Reviewer comments:
Should intra- and inter examiner reliability be published separately. This seems to be the most important part of the current paper.
Our Reply:
Table 3 and table 4 report different results; as table 3 reports inter/intra examiner agreement, while table 4 reports reliability. As described by Kottner & Streiner (J Clin Epidemiol 2011, 64, 701-2.) there is a confusion around reliability and agreement estimation caused by conceptual ambiguities. Agreement points to the question, whether diagnoses, scores, or judgments are identical or similar or the degree to which they differ. In this situation, the absolute degree of measurement error is of interest. Reliability is typically defined as the ratio of variability between scores of the same subjects to the total variability of all scores in the sample.
Reviewer comments:
The paragraph in the results that begins on line 172 is very confusing. Could this be presented in a table format more clearly? A little more information about what these results mean would also be helpful.
Our Reply:
In Introduction and Materials and Methods sections we added a more extensive definition of what misclassification is. Moreover, we tried to change the results section in more clear way.
Reviewer comments:
Could you a priori group the classifications in IDCAS the would represent sound and decayed so that it could be more closely compared to DMFT?
Our Reply:
The aim of the present survey is to compare the methods according their specificities. However, in Table 6 it is possible to compare the methods like as a dichotomy of code (decayed/sound). Regarding the decayed elements for the ICDAS system it is possible to add Dentine, Distinct / Active Cavity and pulp involvement to have a comparison with the single tooth condition related to carious lesions for DMFT.
Reviewer comments:
As written, the discussion and conclusion do not convince me that the comparison is important. Could you add more about what is learned about each classification system from your research and that means for readers?
Our Reply:
We revised the discussion and re-wrote the conclusion section "In conclusion in this research work, while a certain grade of accordance among all the methods was present for the severe caries levels, no accordance was found for the initial (enamel) lesions. From clinical, epidemiological and research prospective both the severity and the activity of a caries lesion are important factors to consider.
The need of a common and standardized language in caries detection is of primary importance when different studies are compared."
Reviewer comments:
Moderate English changes required
Our Reply:
The manuscript was revised by a native English speaker.
Reviewer 2 Report
This study compared several caries assessment methods, including WHO-DMFT, the ICDAS, the CAST and the Nyvad criteria. A lot of effort has been done to collect and analyse the data. My suggestions are as below,
The objectives of the study should be illustrated more clearly as in which aspects are the study comparing. The definition of misclassification should be clearly described. Majority of the conclusion of the study is already known or has been stated in the introduction, such as DMFT is simply but does not take into account of progression of the caries, ICDAS is comprehensive and detailed but time consuming, no need to repeat. What are the new findings or clinical implications of current study?Author Response
Here in detail, the point to point replies to reviewer's comments
Reviewer comment
The objectives of the study should be illustrated more clearly as in which aspects are the study comparing.
Our reply
We deeply modified the introduction and the aim.
Reviewer comment
The definition of misclassification should be clearly described.
Our reply
The definition of misclassification was clarified and added into introduction “In clinical data assessment and analysis, it is important that the variables stating the disease (i.e. Caries) are registered with the least error as possible. Often, the measurements obtained are error prone. When the variables under consideration are categorical, such error is termed misclassification error” (Clin Oral Invest, 2013, 17, 1799–1805).
Reviewer comment
As written, the discussion and conclusion do not convince me that the comparison is important. Could you add more about what is learned about each classification system from your research and that means for readers?
Our reply
We revised the discussion and re-wrote the conclusion section "In conclusion in this research work, while a certain grade of accordance among all the methods was present for the severe caries levels, no accordance was found for the initial (enamel) lesions. From clinical, epidemiological and research prospective both the severity and the activity of a caries lesion are important factors to consider.
The need of a common and standardized language in caries detection is of primary importance when different studies are compared.
Reviewer comment
Moderate English changes required
Our reply
The manuscript was revised by a native English speaker.